# Individual and Combined Antioxidant Activity of Spices and Spice Phenolics

**DOI:** 10.3390/antiox12020308

**Published:** 2023-01-28

**Authors:** Mohammad B. Hossain, Lubna Ahmed, Anna Belen Martin-Diana, Nigel P. Brunton, Catherine Barry-Ryan

**Affiliations:** 1School of Food Science and Environmental Health, Technological University of Dublin, D07 EWV4 Dublin, Ireland; 2Teagasc, Ashtown Food Research Centre, Ashtown, R93 XE12 Carlow, Ireland; 3Agricultural Technological Institute of Castilla and Leon, Government of Castilla and Leon, Finca Zamadueñas, Castilla and Leon, 47071 Valladolid, Spain; 4School of Agriculture and Food Science, University College Dublin, D04 V1W8 Dublin, Ireland

**Keywords:** spices, antioxidants, synergistic, antagonistic, additive, phenolics

## Abstract

The present study investigated the interaction effects (additive, synergistic, and antagonistic) of different groups of spices, their constituent phenolic compounds, and synthetic antioxidants on the total phenol (TP) content and antioxidant activity, as measured by the ferric-reducing antioxidant power (FRAP) of the mixtures. The results showed that there was an additive effect in all the groups studied, except for the group containing turmeric or curcumin. The groups containing turmeric or curcumin showed a moderate synergistic effect. Among the groups of spices, the highest summated TP (50.6 mg GAE/mL) and FRAP (106.2 mg Trolox/mL) values were observed in the group containing clove, cinnamon, pimento, rosemary, oregano, and cardamom. In the case of the groups of pure phenolics, the highest summated TP (364.96 mg GAE/mL) and FRAP (1124.25 mg Trolox/mL) values were observed in the group containing eugenol, acetyl eugenol, caffeic acid, and protocatechuic acid. The summated and combined TP and FRAP values of the samples correlated highly with the correlation coefficients (r^2^) of 0.976 and 0.988, respectively, inferring an additive nature of the interaction effect in most of the groups studied. The interactions of phenolics in mixtures are very complex, being affected by a number of factors, and requires more investigations. The current study will add considerable knowledge to the existing literature to understand the diversity and mechanisms of interactions.

## 1. Introduction

The interest for natural antioxidants for use in foods as a replacement of potentially harmful synthetic antioxidants, such as BHA and BHT, has increased considerably in recent years [1,2]. Natural antioxidants have been shown to have a range of biological properties, for example anticarcinogenic, antimutagenic, antidiabetic, hypolipidemic, and anti-inflammatory, in addition to preventing lipid oxidation in foods [3,4,5,6]. The oxidation of lipids in food not only lowers the nutritional value [7], but is also associated with cell membrane damage, aging, heart disease, and cancer in living organisms [8]. Spices are a well-known source of natural antioxidant and antimicrobial polyphenols, along with their pleasant taste and aroma. Therefore, food industries are increasingly using spices to reformulate foods with higher antioxidant, antimicrobial, and sensory properties. The key antioxidant polyphenols in spices are rosmarinic acid, caffeic acid, carnosic acid, protocatechuic acid, gallic acid, ferulic acid, eugenol, acetyl-eugenol, carnosol, thymol, curcumin, and capsaicin, which are differentially distributed in different spices [9]. However, spices in foods are generally used as mixtures. Therefore, a combination of natural antioxidants occurs in foods and a combination of different antioxidants might act additively, synergistically, and even antagonistically [10]. An additive effect refers to a food combination that provides the sum of the effects of the individual components; a synergistic effect occurs when the effect is greater than the sum of the individual components, and antagonism occurs when the sum of the effects is less than the mathematical sum that would be predicted from the individual components. 

A number of studies have shown the additive [11], synergistic [12,13,14], and antagonistic [15,16] effects on the antioxidant activity of fruits, vegetables, and their processed products. Despite the high antioxidant potential of spices, the information on the interaction effect (additive, synergistic, and antagonistic) of spices and their constituent phenolics on antioxidant activity is limited. Antioxidant activity of individual pure phenolics is predominantly affected by the number and position of OH and OCH_3_ on the phenolic ring. However, the antioxidant activity of combined extracts is a complex output of various other factors, such as ionization, dissociation, concentration, matrix interference, solvent type, intramolecular hydrogen bonds, etc., as reported by Sroka and Cisowski et al. [17], Foti et al. [18], Lucarini and Pedulli [19], Hang et al. [20], and Biela et al. [21]. The aim of the present study was to investigate the interaction effects of the spices and their constituent phenolics on the antioxidant activity of their mixtures. Twenty-one spices, generally used in the ready-meal industry, sixteen pure polyphenols, and five synthetic antioxidants were investigated in different groups for this purpose.

## 2. Materials and Methods 

### 2.1. Samples and Reagents

The dried and ground spices were kindly provided by AllinAll Ingredients Ltd., Dublin, Ireland. All *Lamiaceae* and *Apiaceae* samples, which were the main constituents of different combinations of herbs and spices used in this study, were cultivated in the northern Negev Desert, Israel (Latitude 30 30′ ON, Longitude 34 55′ OE, annual rainfall 12 inches). The herbs were transported to Ireland in premium condition, at 1–3 °C, within 3 days after harvesting. These samples were immediately steamed (120 °C) and air-dried, prior to grinding to <500 µm. Folin–Ciocalteu reagent, sodium acetate anhydrous, ferric chloride hexahydrate, 2,4,6-Tri(2-pyridyl)-s-triazine, 6-Hydroxy-2,5,7,8-tetramethylchroman-2-carboxylic acid, sodium carbonate, butylated hydroxyanisole (BHA), butylated hydroxytoluene (BHT), octyl gallate (OG), propyl gallate (PG), tert-butyl hydroquinone (TBHQ), and the pure phenolics, namely eugenol, acetyl-eugenol, caffeic acid, protocatechuic acid, rosmarinic acid, carnosol, carnosic acid, thymol, curcumin, capsaicin, p-coumaric acid, kaempferol, catechin, gallic acid, ferulic acid, and quercetin were purchased from Sigma-Aldrich, Wicklow, Ireland.

### 2.2. Conventional Solid/Liquid Extraction

Solid/liquid extractions were carried out according to the method of Shan et al. (2005), with slight modifications. Briefly, dried and ground samples (2.5 g) were homogenized for 1 min at 24,000 rpm using an Ultra-Turrax T-25 Tissue homogenizer (Janke and Kunkel, IKA^®^-Labortechnik, Saufen, Germany) in 25 mL of 80% methanol in water (*v*/*v*) at room temperature (~23 °C). The homogenized sample suspension was shaken for overnight with a V400 Multitude Vortexer (Alpha laboratories, North York, Canada) at 1500 rpm at room temperature (≅25 °C). The sample suspension was then centrifuged for 15 min at 2000 g (MSE Mistral 3000i, Sanyo Gallenkamp, Leicestershire, UK) and filtered through 0.45 µm polytetrafluoethylene (PTFE) filters. The extracts were immediately analyzed individually for their total phenol content and antioxidant activity. The combined antioxidant activity was measured after mixing equal volumes of different individual extracts and incubating them for 2 h at room temperature. This led to the dilution of individual extracts. Therefore, the mixture was concentrated under nitrogen to adjust the volume equal to the volume of individual extracts using graduated measuring cylinder. The experiment was performed in two batches, which included three replications of each sample group. The sample groups were selected based on their suitability to be used in popular ready meals such as cottage pie, chicken supreme, and vegetable soups. The groups of the spices and pure phenolics have been listed in Table 1. Food industry often uses synthetic antioxidant to prevent lipid oxidation, while spices are used to enhance taste and aroma. Therefore, 5 different synthetic antioxidants, such as butylated hydroxyanisole (BHA), butylated hydroxytoluene (BHT), octyl gallate (OG), propyl gallate (PG), and tert-butyl hydroquinone (TBHQ), were mixed with the groups of pure phenolics to investigate if there is any interaction between spice phenolics and synthetic antioxidants. The amount of spice powder added per milliliter of solvent (80% methanol) was 100 mg, while an aliquot of 25 mg pure phenolics or synthetic antioxidants was added to the same volume of solvent. Appropriate dilutions were made to achieve the absorbances of the extracts within the standard curve. However, final results were calculated taking the dilutions factor in considerations. 

### 2.3. Determination of Total Phenol (TP)

The total phenolic content was determined using Folin–Ciocalteu reagent (FCR) [22]. The experiment was performed in two batches, which included three replications of each sample and standard. Methanolic gallic acid solutions (10–400 mg/L) were used as standards. In each replicate, 100 µL of the appropriately diluted sample extract, 100 µL methanol, 100 µL FCR, and finally 700 µL Na_2_CO_3_ (20%), were added together and vortexed. The mixture was incubated for 20 min in the dark at room temperature. After incubation, the mixture was centrifuged at 13,000 rpm for 3 min. The absorbance of the supernatant was measured at 735 nm by spectrophotometer (Shimadzu UV-1700, Suzhou Instruments Manufacturing Company Limited, Suzhou, China). The total phenolic content was expressed as gallic acid equivalent (GAE)/mL of the spice extract or pure phenolics.

### 2.4. Ferric Ion-Reducing Antioxidant Power (FRAP) Assay

The FRAP assay was carried out with slight modifications [6]. The FRAP reagent was prepared by mixing 38 mM sodium acetate anhydrous in distilled water pH 3.6, 20 mM FeCl_3_._6_H_2_O in distilled water, and 10 mM 2,4,6-Tri(2-pyridyl)-s-triazine (TPTZ) in 40 mM HCl in a proportion of 10:1:1. This reagent was freshly prepared before each experiment. To each sample, 100 µL of appropriately diluted sample extract and 900 µL of FRAP reagent was added, and the mixture was incubated at 37 °C for 40 min in the dark. In the case of the blank, 100 µL of methanol was added to 900 µL of FRAP reagent. The absorbance of the resulting solution was measured at 593 nm by spectrophotometer. Trolox (6-Hydroxy-2,5,7,8-tetramethylchroman-2-carboxylic acid) (a synthetic antioxidant) at concentrations from 0.1 mM-0.4 mM was used as a reference antioxidant standard. FRAP values were expressed as mg Trolox/mL of the spice extract or pure phenolics.

### 2.5. Quantification of Polyphenols by UHPLC-MS/MS

The polyphenols in the selected fractions were identified and quantified using Waters Acquity (Waters Corporation, Milford, MA, USA) ultra-high performance liquid chromatography coupled with tandem quadrupole mass spectrometry (UHPLC-TQD-MS). The LC separation of the analytes was performed on a Waters Acquity HSS T3 UHPLC column (1.8 μm, 2.1 × 100 mm) using milli-Q^®^ (18 mΩ) (Merck Millipore, Molsheim, France) water (mobile phase A) and acetonitrile:methanol (1:1) containing 0.5% formic acid (mobile phase B). A gradient program of 0–2.5 min 2% B, 2.5–3 min 10% B, 3–7.5 min 15% B, 7.5–8.5 min 35% B, 8.5–9.5 min 98% B, and 9.5–10.0 min 2% B at a flow rate of 0.5 mL/min was used. A multiple reaction monitoring (MRM) approach was taken for the mass spectrometric determination of the polyphenols using argon as collision gas. The parameters for MRM transitions were obtained using the Waters integrated IntellistartTM software (Waters Corp., Milford, MA, USA) (Table 2).

### 2.6. Statistical Analysis

Analysis of variance (ANOVA) was carried out using the software STATGRAPHICS Centurion XVI (Statgraphics Technologies, Inc., Warrenton, VA, USA). ANOVA test was carried out for all experimental runs to determine significant differences among sample groups at α = 0.05 levels. Determination of synergistic, antagonistic, and additive effects were carried out using the following formulae: 

Synergistic effect: TP/FRAP value of the volume adjusted combined extract > sum of the individual extracts (determined by Student’s *t*-test, *p* < 0.05).

Antagonistic effect: TP/FRAP value of the volume adjusted combined extract < sum of the individual extracts (determined by Student’s *t*-test, *p* < 0.05).

Additive effect: TP/FRAP value of the volume adjusted combined extract = sum of the individual extracts (determined by Student’s *t*-test, *p* < 0.05).

## 3. Results and Discussion

### 3.1. Total Phenol Content of Individual and Combined Extracts

The total phenol content of the different groups of spices showed a wide variation. The highest summated total phenol content was observed in group one, containing clove, cinnamon, rosemary, oregano, pimento, and cardamom extracts (Figure 1A). These spices were reported to have a high total phenolic content [9]. On the other hand, the lowest summated total phenol content was observed in group two, where the participating spices were onion, coriander, garlic, ginger, chili, and paprika.

The summated total phenol content in most cases, except group three containing celery, parsley, thyme, basil, sage, and turmeric, did not show any significant difference (*p* < 0.05) when compared to the total phenol content of the combined extracts. This indicated that there was no significant synergistic or antagonistic effect of the different spices within the groups on total phenol content. The results showed the additive effect of the spices on the total phenol content when combined. Similar results were reported [23], which analyzes the combination effects of fruits and vegetables mixtures on the total phenol content. The moderate synergistic effect observed in group three could be attributed to the turmeric extract. To confirm the effect of turmeric, a group (group five) containing all the spices of group three excluding turmeric was formed. In this case, the spices showed an additive effect. This authenticated that turmeric extract might have interacted with one of the remaining five spice extracts to produce synergistic effect on the total phenol content. Similar to the groups of spices, for the groups of pure phenolics, except group eight containing curcumin, the principal component of turmeric did not show any synergistic effect on the total phenol content of the combined solutions of pure phenolics (Figure 1C). Among the groups of different pure phenolics, group six, which contained eugenol, acetyl eugenol, caffeic acid, and protocatechuic acid, showed the highest total phenol content. This could be explained by the high reactivity of eugenol, caffeic acid, and protocatechuic acid with FCR. Group nine containing caffeic acid, gallic acid, ferulic acid, and quercetin also showed a high total phenol content. This group was further analyzed using the LC-MS/MS method to demonstrate if any molecular bonding among phenolics took place, which would result in changes of the masses, ion areas, and ion counts of the constituent molecules. The ion area and ion count of the constituent phenols were recorded for individual solutions (one phenol per solution), and the combined solution had exactly the same concentration. The ion area and ion count of the phenolics in individual solutions were almost identical to those of the combined extracts, indicating no chemical interactions among the phenolics in the combined extract (Table 3, Figure 2). This explains the additive nature of the total phenol contents measured by the Folin– Ciocalteu reagents. In line with these findings, synthetic antioxidants also did not show any interaction with the pure spice phenolics. The mixtures of pure phenolics and synthetic antioxidants showed a cumulative effect on the total phenol content, except for the mixture containing curcumin. Among the synthetic antioxidants tested, the highest total phenol value was observed in the solutions of PG. This was reflected in the total phenol content of the mixtures containing pure phenolics and PG, which showed higher values than those of their respective mixtures containing other synthetic antioxidants. The mixtures of PG were followed by the mixtures of BHA, OG, TBHQ, and BHT, in terms of their total phenol content value (Figure 3, Figure 4 and Figure 5).

### 3.2. Antioxidant Activity of Individual and Combined Extracts

The antioxidant activity of the extracts as measured by the FRAP assay followed the same trend as the total phenol content (Figure 1B). This was evident in the high value of the correlation coefficient (r^2^ = 0.947) between the FRAP and TP results of all the samples tested. In fact, both methods measured the electron donating ability of the antioxidant compounds. The highest antioxidant activity was observed in group one, which contained clove, pimento, cinnamon, rosemary, oregano, and cardamom (Figure 1B). This was expected, as the extracts of clove, pimento, cinnamon, rosemary, and oregano are well-known for their high antioxidant activity. In fact, in an earlier study, these spices were ranked as the top five antioxidant spices among the 30 different spices examined [24]. The antioxidant activity of 3139 food products was investigated [25] using FRAP assay and found that clove had the highest antioxidant activity of all the products tested. The other spices in group 1 were also placed highly on the ranking of antioxidant food products. At the molecular level, the key phenolic compounds of this group are rosmarinic acid and eugenol. Rosmarinic acid in particular showed very strong antioxidant activity among the pure phenolics tested in the current study. In fact, rosmarinic acid showed the highest antioxidant activity among the 16 pure phenolics tested, followed by protocatechuic acid and eugenol. The presence of the four OH groups on the two aromatic rings of caffeic acid and 3,4-dihydroxyphenyllactic acid ester could be attributed to its high antioxidant activity. The high antioxidant activity of protocatechuic acid might be related to its two OH groups at positions two and three the aromatic ring. There was limited information on the relationship between the molecular structure and high antioxidant activity of eugenol. The number and positions of hydroxyl groups in the molecule of eugenol could not explain the high antioxidant activity of eugenol as it had only one hydroxyl group. Other factors such as molecular resonance, bond dissociation enthalpy and/or presence and position of the allyl chain substituted para to the hydroxyl group might have been responsible. Absence of electron withdrawing COOH group and presence of electron donating –OCH_3_ at the ortho position to the hydroxyl group could be associated with its high antioxidant activity. On the other hand, the lowest antioxidant activity was observed in group two, containing onion, coriander, ginger, garlic, chili, and paprika, which were expected as the extracts of these spices showed low antioxidant activity [9,24]. A moderate antioxidant capacity was observed in group three and group four. All the groups of spices, except group three, which contained celery, parsley, thyme, basil, sage, and turmeric, showed an additive effect on the antioxidant activity of the mixture. Similar results were reported by [23], who found a predominantly additive effect of fruits and vegetables mixtures on their antioxidant activity. Group five, which contained all the spices of group three except turmeric, showed an additive effect. This indicated that turmeric was responsible for the synergistic antioxidant activity of group three. The presence of enolic hydroxyl groups in curcumin molecules in organic solvents might have attributed the observed synergistic effects to the combinations containing turmeric and/or curcumin. The synergistic antioxidant effect of curcumin with resveratrol has been reported while analyzing fipronil-induced tissue damage [26]. Among the groups of pure phenolics, group six showed the highest antioxidant activity, which could be related to the high antioxidant activity of the protocatechuic acid (429.27 mg Trolox/mL), eugenol (376.66 mg Trolox/mL), and caffeic acid (315.37 mg Trolox/mL). Group six was followed by group nine in terms of their antioxidant activity (Figure 1B). The major antioxidant contributors of group nine were caffeic acid and gallic acid (348.71 mg Trolox/mL). The high antioxidant activity of gallic acid and caffeic acid could be associated with the presence of three OH groups on ortho positions and two OH groups on para positions of the aromatic rings of these compounds, respectively. Similar to the total phenol content assay, only the mixture containing turmeric extracts showed synergistic effect. A number of studies have investigated the interactions of phenolic compounds in plant extracts or in pure solutions using a diverse set of antioxidant assays, and reported the occurrence of both synergistic and antagonistic interactions [27,28,29,30,31,32,33,34]. The present study observed predominantly an additive interaction, i.e., no interaction among the group of spice extracts and pure phenolic solutions. Hajimehdipoor et al. [28] reported a synergistic interaction between caffeic acid and rosmarinic acid in a binary mixture. However, synergistic interaction was reduced when analyzed in a ternary combination with other phenolic compounds, such as quercetin. Since the present study used a complex mixture of spice extracts and quaternary or more pure phenolics, the interactions in the complex mixture might have been diminished, resulting in an additive effect. 

### 3.3. Correlations of the Antioxidant Activities of Combined and Individual Extracts

The antioxidant phenolic compounds in spices probably had similar chemical characteristics while having no chemical affinity among them. This was probably the reason why the spices and their phenolics did not have a synergistic or antagonistic effect on the antioxidant capacity. Both a synergistic and antagonistic effect would have created a difference between the combined and summated values of the total phenol contents and antioxidant capacities in all the groups studied. A perfect interaction effect would have generated a correlation co-efficient value of one. The correlation coefficients (r^2^) between the summated and combined TP and FRAP values of all the groups studied were 0.976 and 0.988, respectively (Figure 6). In addition, the TP and FRAP values were also highly correlated, with a correlation (r^2^) value of 0.96. Since both the assays follow the same principle of an antioxidant assay of electron transfer-based mechanism, a high degree of correlation in the results was expected. The results of the combination effect of the pure phenolics on antioxidant activity were in agreement with the findings of Heo et al. [35], who reported an additive effect of the mixtures of chlorogenic acid with 11 different phenolics on the antioxidant activity. The mixtures of pure phenolics and the synthetic antioxidants also showed an additive effect, except for the mixture containing curcumin (Figure 3, Figure 4 and Figure 5). An additive effect while investigating the interaction effect of rosemary extract containing rosmarinic acid, carnosic acid, and carnosol with BHA on antioxidant activity was reported [36]. The additive effect observed in most of the groups was reflected in the high level of correlation between the summated and combined TP and FRAP results. 

Strong synergistic or antagonistic effects would have generated considerably low correlation coefficients. Therefore, a moderate synergistic effect observed in the groups containing turmeric and curcumin reduced the correlation coefficients slightly. When the values of the groups containing turmeric and curcumin were excluded from the calculation of correlation, the correlation coefficients between the summated and combined TP and FRAP values increased to 0.998 and 0.999, respectively.

## 4. Conclusions

Extracts of spices, when combined in general, had an additive result in terms of their total phenol content and antioxidant activity, except for turmeric. The well-known health-enhancing spice turmeric attributed a moderate synergistic effect to these combinations. The pure phenolics of spices except for curcumin and the principal phenolic component of turmeric showed similar results. The commonly used synthetic antioxidants also showed an additive effect on the total phenol content and antioxidant activity of the mixtures, excluding turmeric and/or curcumin. Complex matrix interference and a lack of affinity among electron-donating compounds might have been responsible for the largely additive effects observed in the combined extracts and/or solutions. The antioxidant activity of a spice mixture depends on the antioxidant potential of the constituent pure phenolics and their concentration in the spices. Therefore, extracts and/or pure solutions containing highly antioxidant compounds such as rosmarinic acid, protocatechuic acid, eugenol, caffeic acid, and gallic acid showed high combined antioxidant activity.

## Figures and Tables

**Figure 1 antioxidants-12-00308-f001:**
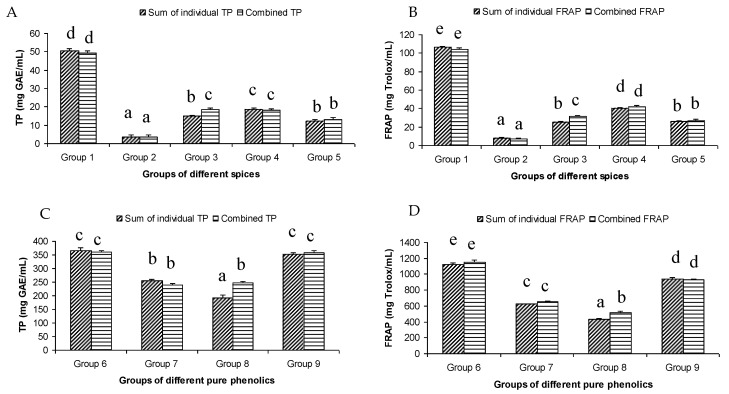
Summated and combined total phenolic (TP) content and ferric-reducing antioxidant power (FRAP) of different groups of spices ((**A**) for TP and (**B**) for FRAP) and pure phenolics ((**C**) for TP and (**D**) for FRAP). Same letters on top of bars in each figure denoted no significant (*p* > 0.05) difference, whereas different letters meant significant difference (*p* < 0.05).

**Figure 2 antioxidants-12-00308-f002:**
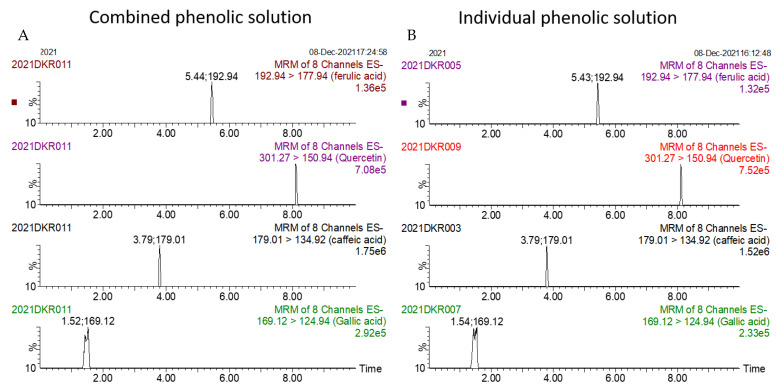
Ion counts of phenolic constituents of combined (**A**) and individual (**B**) extracts as obtained by LC-MS/MS.

**Figure 3 antioxidants-12-00308-f003:**
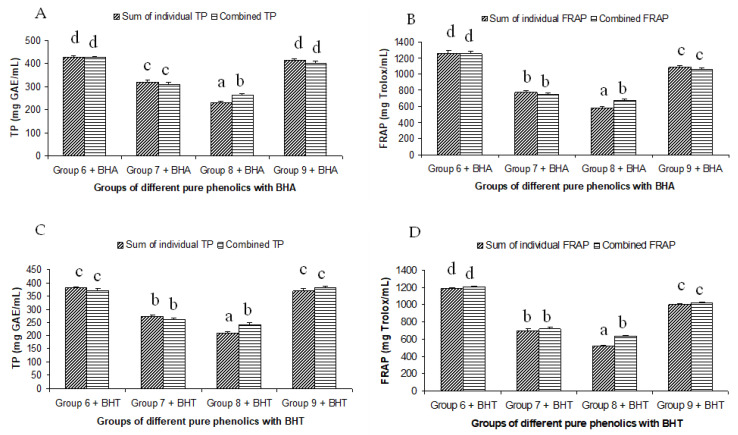
Summated and combined total phenolic (TP) content and ferric-reducing antioxidant power (FRAP) of different groups of pure phenolics with synthetic antioxidants BHA ((**A**) for TP and (**B**) for FRAP) and BHT ((**C**) for TP and (**D**) for FRAP). Same letters on top of bars in each figure denoted no significant (*p* > 0.05) difference, whereas different letters meant significant difference (*p* < 0.05).

**Figure 4 antioxidants-12-00308-f004:**
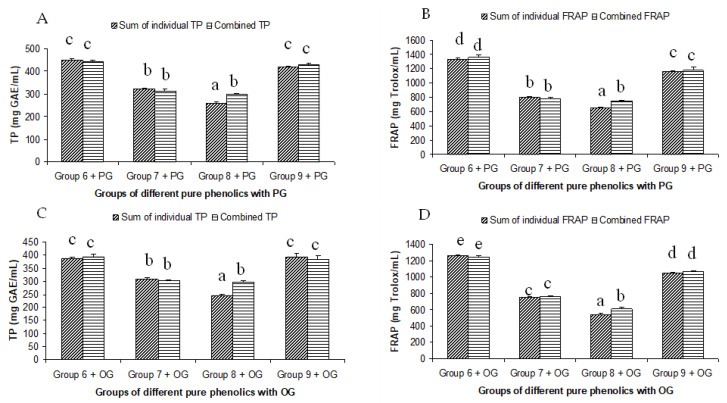
Summated and combined total phenolic (TP) content and ferric-reducing antioxidant power (FRAP) of different groups of pure phenolics with synthetic antioxidants PG ((**A**) for TP and (**B**) for FRAP) and OG ((**C**) for TP and (**D**) for FRAP). Same letters on top of bars in each figure denoted no significant (*p* > 0.05) difference, whereas different letters meant significant difference (*p* < 0.05).

**Figure 5 antioxidants-12-00308-f005:**
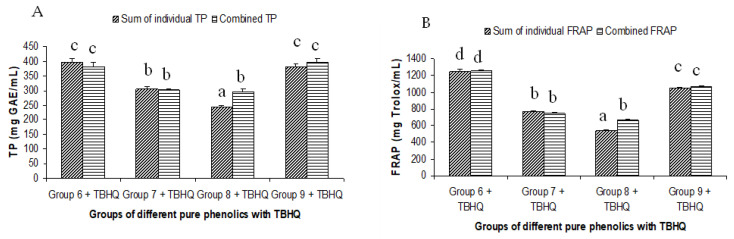
Summated and combined total phenolic (TP) content and ferric-reducing antioxidant power (FRAP) of different groups of pure phenolics with synthetic antioxidants TBHQ ((**A**) for TP and (**B**) for FRAP). Same letters on top of bars in each figure denoted no significant (*p* > 0.05) difference, whereas different letters meant significant difference (*p* < 0.05).

**Figure 6 antioxidants-12-00308-f006:**
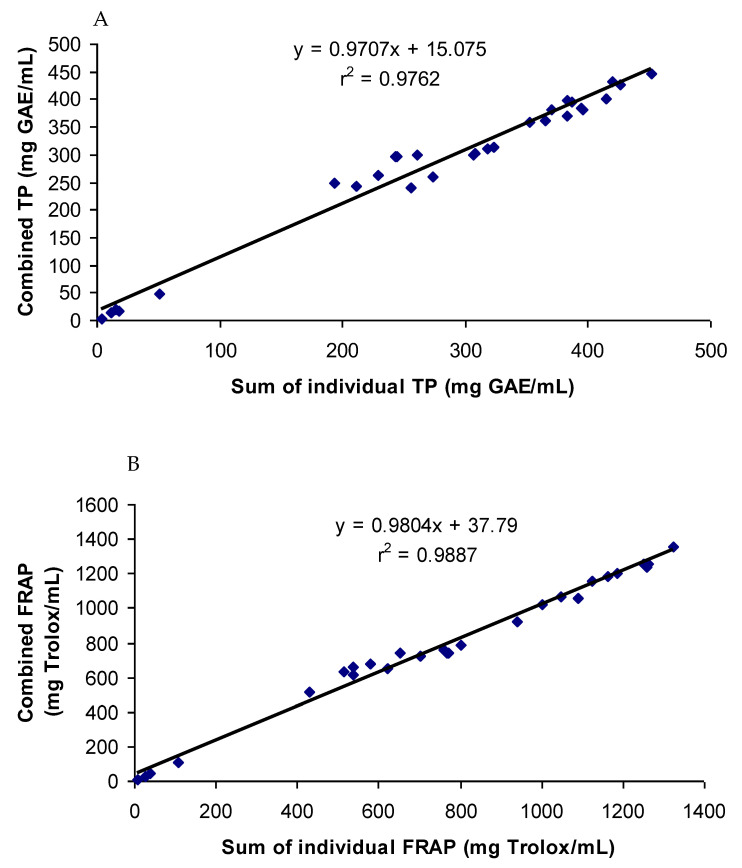
Correlation coefficients between summated and combined TP (**A**) and FRAP (**B**) of all the samples tested.

**Table 1 antioxidants-12-00308-t001:** Composition of the spice and spice phenolics mixture extracted individually and combined at room (~23 °C).

Group No.	Composition	Solute Type	Extraction Solvent	Additional Mixtures
1	Clove, cinnamon, rosemary, oregano, pimento, and cardamom	Mixture of spices	80% methanol	Not applicable
2	Onion, coriander, garlic, ginger, chili, and paprika			
3	Celery, parsley, thyme, basil, sage, and turmeric			
4	Onion, parsley, white pepper, thyme, sage, and cinnamon			
5	Celery, parsley, thyme, basil, and sage			
6	Eugenol, acetyl-eugenol, caffeic acid, and protocatechuic acid	Mixture of spice phenolics		Group 6 plus BHA or BHT, or OG or PG
7	Rosmarinic acid, carnosol, carnosic acid, and thymol			Group 7 plus BHA or BHT, or OG or PG
8	Curcumin, capsaicin, p-coumaric acid, and kaempferol			Group 8 plus BHA or BHT, or OG or PG
9	Caffeic acid, gallic acid, ferulic acid, andquercetin			Group plus BHA or BHT, or OG or PG

**Table 2 antioxidants-12-00308-t002:** The parameters for MRM transitions.

Phenolic Compounds	Molecular Formula	Retention Time (min)	MRM (*m*/*z*)	Cone Voltage (V)	Collision Energy (eV)
Caffeic acid	C_9_H_8_O_4_	3.79	179.01 > 134.92	25	18
Gallic acid	C_7_H_6_O_5_	1.52	169.12 > 124.94	20	16
Quercetin	C_15_H_10_O_7_	8.01	301.27 > 150.94	33	24
Ferulic acid	C_10_H_10_O_4_	5.44	192.94 > 177.94	25	20

**Table 3 antioxidants-12-00308-t003:** Comparison of areas of the peak detected in combined and individual extracts.

Phenolic Compounds	Combined Extract	Individual Extract
	Area ± SD	
Caffeic acid	82,601 ± 4354 d *	83,273 ± 4669 d
Gallic acid	41,555 ± 1464 c	41,928 ± 1763 c
Quercetin	35,545 ± 1119 b	35,382 ± 533 b
Ferulic acid	336 ± 24 a	321 ± 44 a

* Same letters denote the two value compared are not significantly different (*p* > 0.05).

## Data Availability

Data is contained within the article.

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
