# Peer review of "Individual and Combined Antioxidant Activity of Spices and Spice Phenolics"

_antioxidants, 2023, doi:10.3390/antiox12020308_

Round 1
Reviewer 1 Report
The text is prepared quite carelessly, it needs correction.
E.g. line 92 – it is “group group”, should be “group”
Line 113 – it is “Na2CO3”, should be “Na2CO3”
Genus names of Lamiaceae and Apiaceae should be capitalized and italicized.
The charts are illegible, the size of the standard deviation is difficult to assess.
The two drawings are labeled Figure 5.
Figure 5. Ion count of phenolic constituents of combined (A) and individual (B) extracts as obtained by LC-MS/MS.
Figure 5. Correlation coefficients between summated and combined TP (A) and FRAP (B) of all the sample tested.
The chapter on methodology should be supplemented. It is not entirely clear how the mixtures of individual groups 1-9 were prepared.
Lack of justification and commentary in the text of Figure 5 (Ion count….) and Table 2, the discussion of the results is incomplete and requires further development. Conclusions require clarification, they are too general.
Author Response
The authors appreciate and acknowledge the valuable comments, corrections and suggestions of the esteemed reviewers. The corrections have been highlighted with yellow background. The following corrections have been made as per the reviewers’ instructions:
The text is prepared quite carelessly, it needs correction.
- g. line 92 – it is “group group”, should be “group”
Author’s response: Corrected, now in line 97
- Line 113 – it is “Na2CO3”, should be “Na2CO3”
Author’s response: Corrected, now in line 120
- Genus names of Lamiaceaeand Apiaceae should be capitalized and italicized.
Author’s response: Corrected, now in line 68
- The charts are illegible, the size of the standard deviation is difficult to assess.
Author’s response: Thanks for your invaluable comments. Since the two antioxidant assays are in vitro, the variation among the results are minimal. Hence the standard deviation is so small. The high antioxidant values which were much larger than the standard deviation, made the standard deviation bar visually tiny. However, letter annotation on top of the each bar expressed the statistical difference, if there was any.
- The two drawings are labeled Figure 5.
Figure 5. Ion count of phenolic constituents of combined (A) and individual (B) extracts as obtained by LC-MS/MS.
Figure 5. Correlation coefficients between summated and combined TP (A) and FRAP (B) of all the sample tested.
Author’s response: Figure 5 has been repositioned as Figure 2. Second Figure 5 has been corrected as Figure 6. Now in line 219 and 320, respectively.
- The chapter on methodology should be supplemented. It is not entirely clear how the mixtures of individual groups 1-9 were prepared.
Author’s response: Table 1, outlining the compositions of group 1 to 9 has been added.
- Lack of justification and commentary in the text of Figure 5 (Ion count….) and Table 2, the discussion of the results is incomplete and requires further development. Conclusions require clarification, they are too general.
Author’s response: Justification of Figure 5 (Now Figure 2) and Table 2 (Now Table 3) has been added in line 196-198. Discussion has been improved. Now lines 196-198, 250-265, 283-285, 287-296, 302-305 and 307-310 have been added. Conclusion has been improved as well. Now lines 329-331, 335-337, 339-341 have been added.
Reviewer 2 Report
This manuscript describes an interesting investigation that provides new data on the interaction effect of different groups of spices, their constituent phenolic compounds and synthetic antioxidants on the total phenol content and antioxidant activity of the mixtures.
The issues addressed are relevant in the field and aim to fill some of the existing gaps regarding the additive, synergistic and antagonistic effect of investigated spices, their phenolic compounds and synthetic antioxidants on TP content and antioxidant activity of the mixtures.
This study is very promising, however there are some aspects that could improve it significantly:
· I am aware of the word limit in the abstract, but I think it is important to add one or two relevant sentences to highlight the impact of this study on the current state of knowledge.
· The introduction gives an overview of current research on the subject, but is too brief. In order to create a solid context on the topic, this part should have been more substantial. Some improvements could be made in this regard.
· The methodology is well addressed and provided in sufficient detail. In section 2.2, please prepare a table with the groups (1-9) of spices and pure phenolics investigated in this study.
· In Figures 1-4 from section 3.1. Total phenol content of individual and combined extracts, I recommend you to present only the TC data, not the FRAP value data. The FRAP value data will be included in other figures that will be presented separately in section 3.2. Antioxidant activity of individual and combined extracts.
· I suggest to extend the discussion by addressing the correlations between TP and FRAP value in another point, such as 3.3.
· Please, improve the conclusions, making them more consistent, according to the obtained results. The authors did not emphasize the innovative aspect of their work that support the added value; it should be added to the conclusion.
· The references cited are relevant to this research topic, but too few. Please improve the references list by adding more studies related to this topic.

Author Response
The authors would like to thank the esteemed reviewers for their constructive comments and corrections.
Reviewer 2
This manuscript describes an interesting investigation that provides new data on the interaction effect of different groups of spices, their constituent phenolic compounds and synthetic antioxidants on the total phenol content and antioxidant activity of the mixtures.
The issues addressed are relevant in the field and aim to fill some of the existing gaps regarding the additive, synergistic and antagonistic effect of investigated spices, their phenolic compounds and synthetic antioxidants on TP content and antioxidant activity of the mixtures.
This study is very promising, however there are some aspects that could improve it significantly:
- I am aware of the word limit in the abstract, but I think it is important to add one or two relevant sentences to highlight the impact of this study on the current state of knowledge.
Author’s response: Added, now in line 25-27
- The introduction gives an overview of current research on the subject, but is too brief. In order to create a solid context on the topic, this part should have been more substantial. Some improvements could be made in this regard.
Author’s response: Authors would like to thank the reviewer for his/her valuable comments. More information has been added to the introduction to improve the context of the present study. Now in line 55-60.
- The methodology is well addressed and provided in sufficient detail.In section 2.2, please prepare a table with the groups (1-9) of spices and pure phenolics investigated in this study.
Author’s response: Spice and spice phenolics groups (1-9) has been listed in Table 1
- In Figures 1-4 from section 3.1. Total phenol content of individual and combined extracts, I recommend you to present only the TC data, not the FRAP value data. The FRAP value data will be included in other figures that will be presented separately in section 3.2. Antioxidant activity of individual and combined extracts.
Author’s response: The authors appreciate the suggestions to clarify the results by splitting TP and FRAP figures. However, the authors view is that parallel positioning of TP and FRAP graphs would help the reader to have a comparison between the TP and FRAP values.
- I suggest to extend the discussion by addressing the correlations between TP and FRAP value in another point, such as 3.3.
Author’s response: Discussions on correlations have been extended and moved to section 3.3
- Please, improve the conclusions, making them more consistent, according to the obtained results. The authors did not emphasize the innovative aspect of their work that support the added value; it should be added to the conclusion.
Author’s response: Conclusion has been improved. Now in line 329-331, 335-337 and 339-341
- The references cited are relevant to this research topic, but too few. Please improve the references list by adding more studies related to this topic.
Author’s response: New references 18-21 and 27-34 have been added.
Round 2
Reviewer 2 Report
All suggested recommendations have been well addressed by the authors. The paper has been improved accordingly and can therefore be accepted for publication in this form.